# Ecological divergence of wild birds drives avian influenza spillover and global spread

**Nichola J. Hill**[ID][1]*, **Mary Anne Bishop**[ID][2], **Nídia S. Trovão**[3], **Katherine M. Ineson**[ID][4], **Anne L. Schaefer**[ID][2], **Wendy B. Puryear**[ID][5], **Katherine Zhou**[6], **Alexa D. Foss**[ID][5], **Daniel E. Clark**[ID][7], **Kenneth G. MacKenzie**[ID][7], **Jonathon D. Gass, Jr.**[ID][5], **Laura K. Borkenhagen**[ID][5], **Jeffrey S. Hall**[8], **Jonathan A. Runstadler**[5]

**1** Department of Biology, University of Massachusetts, Boston, Massachusetts, United States of America, **2** Prince William Sound Science Center, Cordova, Alaska, United States of America, **3** Division of International Epidemiology and Population Studies, Fogarty International Center, National Institutes of Health, Bethesda, Maryland, United States of America, **4** U.S. Fish and Wildlife Service, Hadley, Massachusetts, United States of America, **5** Department of Infectious Disease and Global Health, Cummings School of Veterinary Medicine Tufts University, North Grafton, Massachusetts, United States of America, **6** College of Veterinary Medicine, Cornell University, Ithaca, New York, United States of America, **7** Division of Water Supply Protection, Massachusetts Department of Conservation and Recreation, West Boylston, Massachusetts, United States of America, **8** U.S. Geological Survey, National Wildlife Health Center, Madison, Wisconsin, United States of America

* nichola.hill@umb.edu

**Data Availability Statement:** Genetic sequences from influenza A virus surveillance from Alaska and Massachusetts are deposited in GenBank (https://

## Abstract

The diversity of influenza A viruses (IAV) is primarily hosted by two highly divergent avian orders: Anseriformes (ducks, swans and geese) and Charadriiformes (gulls, terns and shorebirds). Studies of IAV have historically focused on Anseriformes, specifically dabbling ducks, overlooking the diversity of hosts in nature, including gull and goose species that have successfully adapted to human habitats. This study sought to address this imbalance by characterizing spillover dynamics and global transmission patterns of IAV over 10 years at greater taxonomic resolution than previously considered. Furthermore, the circulation of viral subtypes in birds that are either host-adapted (low pathogenic H13, H16) or host-generalist (highly pathogenic avian influenza—HPAI H5) provided a unique opportunity to test and extend models of viral evolution. Using Bayesian phylodynamic modelling we uncovered a complex transmission network that relied on ecologically divergent bird hosts. The generalist subtype, HPAI H5 was driven largely by wild geese and swans that acted as a source for wild ducks, gulls, land birds, and domestic geese. Gulls were responsible for moving HPAI H5 more rapidly than any other host, a finding that may reflect their long-distance, pelagic movements and their immuno-naïve status against this subtype. Wild ducks, long viewed as primary hosts for spillover, occupied an optimal space for viral transmission, contributing to geographic expansion and rapid dispersal of HPAI H5. Evidence of inter-hemispheric dispersal via both the Pacific and Atlantic Rims was detected, supporting surveillance at high latitudes along continental margins to achieve early detection. Both neutral (geographic expansion) and non-neutral (antigenic selection) evolutionary processes were found to shape subtype evolution which manifested as unique geographic hotspots for each subtype at the global scale. This study reveals how a diversity of avian hosts contribute to

www.ncbi.nlm.nih.gov/genbank) under accession
numbers listed in S1 and S2 Data.

**Funding:** Funding for this project was provided by
the NIAID Centers of Excellence for Influenza
Research and Surveillance (HHSN272201400008C
(JR) & HHSN272201400006C (JH)) and the North
Pacific Research Board (project no. 1411 (NH, MB,
JR)). The funders had no role in study design, data
collection and analysis, decision to publish, or
preparation of the manuscript.

**Competing interests:** The authors have declared
that no competing interests exist.

viral spread and spillover with the potential to improve surveillance in an era of rapid global
change.

## Author summary

Our study provides novel insights into the biology of influenza A virus (IAV), which is
timely in view of the unusually large number of animal and human cases of highly patho-
genic avian influenza (HPAI) H5 across Europe, Asia, Africa and North America. Cur-
rently we face challenges with predicting how the avian reservoir will influence IAV
spread because the mechanisms by which different subtypes disperse are not well under-
stood. Our study sought to address this knowledge gap by systematically comparing the
evolutionary dynamics that drive IAV transmission across subtypes and bird hosts with
the goal of identifying spillover pathways at the wild-domestic interface. By analyzing the
evolution of IAV over 10 years at greater taxonomic resolution than previously consid-
ered, we uncovered a complex transmission network that relied on ecologically divergent
bird hosts. Domestic birds were responsible for slow but steady range expansion of HPAI
H5, while wild birds such as geese, swans, gulls and ducks contibuted to rapid but episodic
dispersal via uniquely different pathways. By assessing how virus-host systems are cou-
pled, findings from this study have the potential to refine and enhance global surveillance
and outbreak prediction.

## Introduction

Influenza A viruses (IAVs) are adapted to infect a wide array of hosts, spanning divergent taxo-
nomic groups and hundreds of birds and mammal species, including humans. Despite an
expansive host range, the greatest viral diversity among IAVs is attributed to persistent circula-
tion in aquatic birds, within which 16 of the 18 known hemagglutinin (HA) subtypes and 9 of
the 11 known neuraminidase (NA) subtypes have been identified [1,2]. The viral diversity rep-
resented within aquatic birds is unparalleled by any other host group and is primarily hosted
by two highly divergent avian orders: Anseriformes (ducks, swans and geese) and Charadrii-
formes (gulls, terns and shorebirds). These two avian orders are highly diverged on the avian
phylogeny, with Anseriformes estimated to be an older, ancestral lineage (97.6 million years)
while Charadriiformes are a more recently evolved lineage (79.5 million years) based on multi-
locus phylogenetic dating [3]. The taxonomic differences between these two avian orders may
provide insights into the evolutionary strategies of IAV when exposed to either a closely or dis-
tantly related host. IAV surveillance has historically focused on Anseriformes, specifically dab-
bling ducks. Far less is known about the role of Charadriiformes in facilitating viral
transmission between species and global spread of IAV. Understanding how transmission
dynamics differ between avian reservoirs is of central importance to predicting spillover to
domestic animals and humans, a question with growing relevance in view of the expanding
human-animal interface.

Gulls are one of the most abundant and widespread IAV hosts, and unlike other avian res-
ervoirs, their distribution and abundance is more strongly linked to human activities such as
farming, fishing and urbanization [4,5]. Mounting evidence indicates that global gull popula-
tions are in flux due to changes in the marine ecosystem [6,7]. Anthropogenic pressures on the
natural food web are driving many bird species to local or regional extirpation [8], but in the

case of phenotypically plastic taxa such as many gulls, a switch to anthropogenic sources of food has been documented [4,9]. It is well documented that gulls are primarily infected with two HA subtypes; H13 and H16 [10,11], yet global surveillance has shown that gulls are permissive hosts for all 16 HA subtypes hosted by wild birds [12,13]. The H13 and H16 subtypes are hypothesized to have co-evolved with Charadriiformes, consistent with their predominance in gulls [12,13] and the species-specific molecular markers present in these subtypes [14,15]. A consistent finding of surveillance studies conducted at local or regional-scales has been that gulls play a key role in the inter-hemispheric spread of IAV [16,17] attributed to their long-distance pelagic movements [18,19]. There is a pressing need to compare the roles of avian reservoirs, using longer term, globally distributed data to account for variation in IAV ecology. Due to perturbations to marine ecosystems, this is especially relevant to studying gulls, and the goal of defining their contribution to the mixing, distribution and dispersal of IAV between global regions.

In contrast to the host-adapted H13 and H16, the HPAI H5 subtype is known to have a broad host range encompassing Anseriformes, Galliformes and Charadriiformes, as well as many non-aquatic birds such as passerines and raptors. The global expansion of the HPAI H5 subtype, originating in domestic geese in 1996 in Guandong, China (GsGD lineage) [20] represents a phenotype of IAV with distinctly different evolutionary dynamics from H13 and H16. Viral descendants of A/goose/Guangdong/1/1996 belonging to the HPAI H5 subtype have achieved persistent circulation in domestic poultry and wild ducks with periodic spillover to mammals [21–23]. While evidence suggests that transmission between domestic and wild Anseriformes is key to endemic circulation of HPAI H5 in Asia [24] and Africa [25], the role of gulls as a ubiquitous and globally distributed host for HPAI H5 is unclear. Experimental studies indicate that gulls are a permissive host for multiple lineages of HPAI H5 [26,27], a finding that agrees with consistent detection of this subtype in gulls across Asia and Africa where the virus has become established. Whether gulls play a role in the maintenance of HPAI H5, or they represent a spillover host in which transmission is transient is an important question warranting further investigation. In view of the potential for gulls to undertake long-distance pelagic movements and contribute to inter-continental exchange of the virus, this question has relevance for understanding and predicting the global spread of HPAI H5.

Predicting how certain species will influence viral spread remains fraught because the evolutionary dynamics that drive transmission of subtypes have not been compared systematically across species. The circulation of multiple subtypes in wild birds that are either host-associated (H13, H16) or more host-generalist (HPAI H5) provides a unique opportunity to test and extend models of IAV evolution. The overall objective of this study was to describe the mechanisms by which different subtypes disperse in the same host. Firstly, we sought to explore the evolutionary dynamics of H13, H16 and HPAI H5 to evaluate how they overlapped or differed in their global distribution, rates of dispersal and whether competitive interactions occur between multiple subtypes at the local scale. Secondly, we sought to investigate the specific roles of Anseriformes and Charadriiformes in the inter-hemispheric transmission of these three subtypes. Thirdly, we sought to characterize how avian hosts contribute to wild-domestic bird spillover at taxonomic scales much lower (i.e. family and genus) than previously considered. We tested the paradigm that co-evolution between virus and host is the primary factor governing transmission dynamics at the global scale, and that phylogenetically unrelated taxa play a marginal role in amplification that hinges instead on host abundance. Findings from this study have the potential to refine and enhance global surveillance and outbreak prediction by addressing how virus-host systems are coupled.

## Results

### Evolutionary trajectories are unique for different subtypes

Wild birds were live-caught and sampled across North America between 2008 and 2018 and the 888 influenza sequences generated (including H13: n = 23 sequences and H16: n = 58 sequences) were compared with global reference sequences for the three subtypes. The HA gene phylogeny showed markedly different evolutionary dynamics for each of the three subtypes: H13, H16 and HPAI H5. Molecular dating of the gull-associated subtypes revealed persistence of H16 since approximately 1969 (95% HPD: May 1966 –April 1972) and of H13 since approximately 1973 (95% highest posterior density, HPD: December 1971–June 1974). This suggested a similar time of origin for the two gull-associated subtypes. In comparison, HPAI H5 diversified rapidly over a much shorter time frame since 1996 (95% HPD: January 1996 – December 1996).

For H13, viruses belonged to three major clades that continue to be detected across a wide geographic range including all hemispheres (S1A Text). Concurrent circulation of multiple clades of H13 contrasts with the evolution of H16. For H16, most circulating viruses belong to a single dominant clade that has persisted via localised transmission over a smaller geographic area for successive years (S1B Text). The pattern of endemic circulation is demonstrated in Alaska where the same lineage has persisted for approximately a decade since 2009 (95% HPD: November 2009–June 2010). The mean evolutionary rate (or substitution rate) of H13 was estimated to be $5.81 \times 10^{-3}$ substitutions per site per year (95% HPD: $5.36–6.29 \times 10^{-3}$), faster than H16 for which we estimated a mean evolutionary rate of $4.61 \times 10^{-3}$ (95% HPD: $4.12–5.15 \times 10^{-3}$). These results indicate that the evolutionary dynamics of IAV subtypes can be unique and distinct even when hosted by the same avian reservoir, such as H13 and H16 in gulls.

The evolution of HPAI H5 indicates extensive diversification over time (S1C Text) consistent with the known clade designations determined by the WHO/OIE [28] resulting in the most recent clade 2.3.4.4 (95% HPD: February 2009 –April 2011) that expanded into continental Europe, Asia and Africa and a rare introduction event into North America via the Pacific-Central flyways in approximately 2014 (95% HPD: November 2013 –May 2014). Most lineages predominately circulate in East and Central Asia with the exception of clade 2.2 that since 2005 (95% HPD: November 2004 –July 2005) has become endemic in the Black Sea-Mediterranean and Africa (S1C Text). Diversification of clade 2.3 into multiple co-circulating sub-clades has been associated with a geographic range spanning Eurasia but with no evidence of occurring along the Atlantic seaboard. We estimated that HPAI H5 evolved at a rate of $5.65 \times 10^{-3}$ (95% HPD: $5.21–6.15 \times 10^{-3}$), comparable to the evolutionary rates of H13 and H16.

### Hotspots rarely overlap for subtypes adapted to the same host

We reconstructed the global phylogeography of the three subtypes: H13, H16 and HPAI H5 using a discrete trait model based on 11 geoclusters of the globe. Two geographic areas emerged as major sources of H13: the Mississippi-Atlantic flyway in the US (26.0% Markov jumps) and the Black Sea-Mediterranean region in Eurasia (22.0% Markov jumps: Fig 1A). The Mississippi-Atlantic acted as a source of spread for regions north (North Atlantic) and south (South America). The Black Sea-Mediterranean acted as a source of H13 for continental Europe and East Asia. Using Markov rewards to estimate the proportion of time H13 spent in each region revealed that these viruses spent comparable amounts of time across regions. H13 spent most time in the Mississippi-Atlantic (24.2%) followed by the Black Sea-Mediterranean (15.5%), continental Europe (12.3%), South America (11.3%) and East Asia (10.3%). These viruses spent least amount of time in Alaska and the Pacific-Central flyway of the US. H13

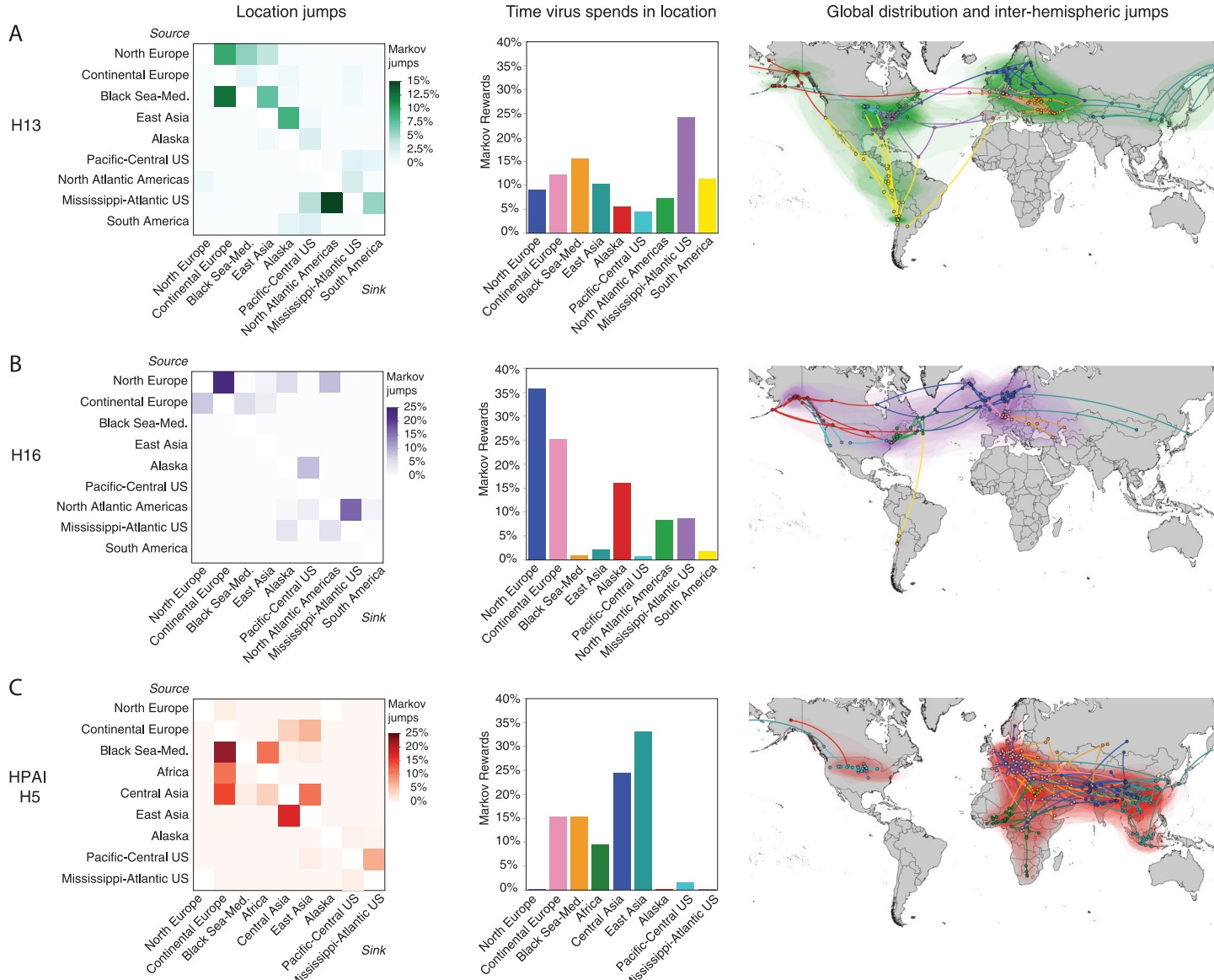

**Fig 1. The contrasting pattern of geographic jumps, hotspots and global distribution of influenza A viral subtypes.** The heatmaps: A) H13 (green), B) H16 (purple) and C) highly pathogenic avian influenza H5 (red), indicate the frequency of transitions between locations estimated using a discrete trait phylogenetic model and the number of location transitions was determined using Markov jumps. The proportion of time the virus spends in each location (center panel) is indicated by the bar charts with each subtype showing a uniquely different geographic distribution. The global distribution of each subtype is depicted by shading on the maps (right panel). Lines indicate the route of inter-hemispheric virus flow and represent the branching pattern of internal and external nodes of the underlying phylogenetic tree. Basemap made with Natural Earth (https://www.naturalearthdata.com).

transmission between the eastern and western hemispheres was rare with the exception of East Asia acting as a repeated source of viruses for Alaska, highlighting this region of connectivity across the Pacific Ocean.

For H16, high latitude regions including Northern Europe (41.0% Markov jumps) and to a lesser degree, the North Atlantic coast of the Americas (23% Markov jumps) were important sources of virus (Fig 1B). Northern and continental Europe were strongly connected (24.1% Markov jumps), as were the North Atlantic and Mississippi-Atlantic flyway of the US (16.8%

Markov jumps), indicating localized patterns of H16 spread. Alaska was also identified as a source of H16 for locations further south along the Pacific and Central flyways of the US. Estimation of the time H16 spent in each region using Markov rewards indicated a pattern highly skewed towards a few locations: Northern and continental Europe (35.9% and 25.2%) and Alaska (16.0%). H16 spent the least amount of time in the Black Sea-Mediterranean and South America. H16 transmission between hemispheres occurred infrequently via Northern Europe into the North Atlantic coast of the Americas, and inter-hemispheric transmission was otherwise limited.

## Geographic distribution of HPAI associated with proximity to endemic hotspots

HPAI H5 revealed a geographic distribution centred in the eastern hemisphere with transmission primarily seeded by the Black Sea-Mediterranean (29.45% Markov jumps), East Asia (17.77% Markov jumps) and Central Asia (17.37% Markov jumps) (Fig 1C). The Black Sea-Mediterranean, Africa and Central Asia were important sources of HPAI H5 for continental Europe, while further transmission to Northern Europe was rarely detected. Viral flow from East to Central Asia was strongly supported by the large number of Markov jumps estimated (17.5%). Consistent with these results, analysis of time spent by HPAI H5 in each geographic location indicated East Asia was the primary site of circulation (33.1%). Furthermore, the amount of time spent by these viruses in each location was based on proximity to East Asia; i.e. Central Asia (24.5%), Black Sea-Mediterranean (15.3%) and Africa (9.5%). Transmission between the eastern and western hemispheres was infrequent, but also appeared to be dictated by proximity to East Asia due to evidence of the HPAI H5 circulating primarily along the Pacific-Central flyways of the US, compared to the Atlantic coast.

## Dispersal between hemispheres occurs via the Pacific and Atlantic Rim

Transmission between North America and Eurasia was consistently rare for all three subtypes, relative to transmission within each region. For H13, viral migration between East Asia and Alaska indicated spread via the Pacific Rim (Fig 1A). A similar route of introduction into North America was estimated for HPAI H5, whereby East Asia was identified as a source for the incursion of virus into the Pacific-Central flyways (Fig 1C). For H16, a different route of introduction into North America was identified involving the Atlantic Rim. Northern Europe acted as a viral source for the Northern Atlantic seaboard of the Americas—a pair of locations in proximity (Fig 1B). In summary, both the Pacific and Atlantic Rims acted as gateways for the introduction of IAVs into North America with a correlation between source and sink locations that are in geographic proximity along the continental margins.

## Competition and co-existence underlies subtype distribution

We focused on Alaska to address how multiple subtypes interact and behave when co-circulating in the same population. The H16 subtype dominated in gulls that migrate and breed in Cordova, Alaska as evidenced by detection of H16 during each of the 10 years of the study (Fig 2). H16 consisted of two geographically distinct clades (Northern Europe and Mississippi-Atlantic US). The temporal distribution of these H16 clades based on multiple components analysis (MCA) indicated that they rarely co-circulated in Cordova, Alaska (S2 Text), suggestive of competitive exclusion occurring between lineages. Multiple introductions of H13 from East Asia and to a lesser degree South America were also observed but persisted for only 1–2 years before they were no longer detected in the gull population. Temporal overlap between H13 from East Asia and H16 from Mississippi-Atlantic US based on MCA, suggested co-circulation and a lack of competition between these two geographically distinct lineages.

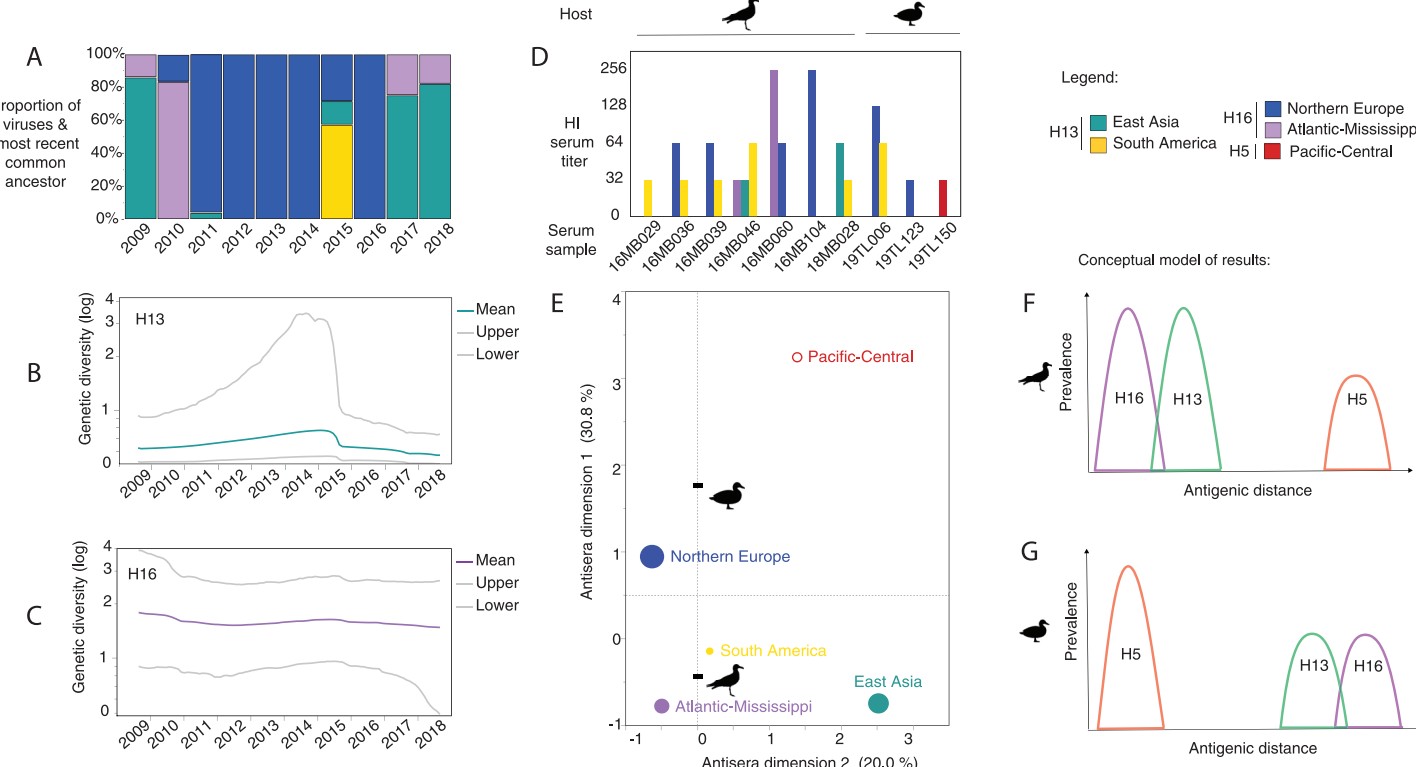

**Fig 2. The occurrence of influenza A viral subtypes over a 10-year period in Alaska reflect neutral and non-neutral evolutionary processes.** A) The temporal pattern consists of two H16 clades (blue: Northern Europe, and purple: Atlantic-Mississippi) that alternate, with introductions of H13 from East Asia (green) and South America (yellow). Geographic origins were determined by most recent common ancestor (MRCA) phylodynamic analysis. B) Genetic diversity of H13 through time C) relative to H16. D) Cross-reactivity of anti-influenza antibodies from Cordova gulls and Minto Flats ducks determined by hemagglutination inhibition (HI) assay. Many antibodies against H13 and H16 were cross-reactive, while few sera recognized H5 low pathogenic virus. E) Antigenic properties of H13, H16 and H5 clades mapped onto two-dimensional space. Virus clades positioned centrally (i.e. H13 from South America) were more cross-reactive than viruses located at the edges (i.e. H5 from Pacific-Central, H13 from East Asia). Circle markers (solid) are size-scaled according to the prevalence of each clade in the Cordova gull population, or their absence (outline). F) Conceptual diagram of the antigenic distance between subtypes relative to prevalence. Gulls are proposed as the reservoir host for H13 and H16, which are antigenically related and overlap resulting in cross-reactivity and competition between H13 and H16. G) Conversely, ducks (and geese and swans) are proposed as the reservoir for highly pathogenic avian influenza H5. No antigenic overlap occurs between HPAI H5, H13 and H16, such that gulls are largely immuno-naïve to HPAI H5. Animal silhouettes from http://phylopic.org.

The genetic diversity of H13 circulating in Cordova, Alaska was low compared to H16, but rapidly accrued with the introduction of the South American clade, as inferred from Bayesian skyride plots that showed changes in viral effective population size over time (Fig 2B and 2C). After introduction of the South American clade, the genetic diversity of H13 markedly declined within one year indicating local extirpation or exclusion of this clade from the viral population. In contrast the diversity of H16 was two-fold higher and relatively stable over time (Fig 2C). We found that despite turnover of the two H16 clades (Northern Europe and Atlantic-Mississippi) over time, the genetic diversity did not increase or diminish suggesting they were genetically equivalent and occupied similar spaces in the viral fitness landscape defined by gulls in Cordova, Alaska.

## Frequent reassortment between prevailing subtypes

To evaluate the degree of overlap and interaction between H13 and H16, we estimated rates of reassortment by quantifying transitions of the PB2 internal segment between subtypes (S3 Text). H16 was a frequent source (or "donor") of PB2 for the H13 subtype infecting the

Cordova gull population, indicating reassortment and competition between subtypes at the individual host level. Conversely, H13 also donated the PB2 segment to H16, but at a rate with lower statistical support (S3 Text). Other HA subtypes that reassorted with H16 included the H1, H2, H5 and H6 group, while H13 reassorted with the H8, H9, H12 group. However, to a large degree the H13 and H16 subtypes showed limited reassortment with other HA subtypes, which primarily swapped PB2 internal segments among each other as a distinct and separate viral pool.

## Subtypes evolve by antigenic selection and geographic structuring

Results from serological testing of Alaskan gulls and ducks also supported interaction between H13 and H16 with evidence of antigenic cross-reactivity between the two subtypes depending on geographic lineage. For example, sera against an H16 clade from Northern Europe was strongly cross-reactive with an H13 clade originating from South America (Fig 2D). Additionally, sera against the H13 clade from South America recognized two other virus clades H13 (from East Asia) and H16 (from the Atlantic-Mississippi) and was therefore the most broadly cross-reactive strain according to its central position in the MCA map (Fig 2E). This result implied antigenic overlap may have caused the low levels of H13 from South America detected in the study population (one out of 10 years). In contrast, the H16 clade from Northern Europe predominated for 7 out of 10 years (Fig 2A) and was not cross-reactive against H13 from East Asia (Fig 2E), the second most common clade in the gull population. The two most common viral clades never temporally overlapped. The sera results indicate that geographic structuring rather than antigenic overlap was responsible for the lack of co-circulation observed.

Unlike the gull-associated subtypes of H13 and H16, the H5 subtype (LPAI) was antigenically unrelated and did not cross-react with any of the other viruses based on the serological analysis. Sera from only one duck recognized the H5 virus and at a relatively low titer of 1:32 hemagglutinating units (Fig 2D). The H5 virus occupied a distant position at the edges of the MCA map that confirmed lack of antigenic overlap with other viruses tested (Fig 2E). Ducks as a host group generated sera that rarely recognized H13 and H16, relative to gulls. Two duck sera recognized the H16 clade from North America, and one recognized the H13 clade from South America (Fig 2D). These findings concur with the view that ducks are largely immuno-naïve against gull-associated viruses, and likewise, gulls are largely immuno-naïve against duck-associated viruses. Overall, this supports that host barriers to virus transmission exist between gulls and ducks, which can subsequently cause immunologically naïve populations with heightened susceptibility to antigenically distinct viruses (Fig 2F and 2G).

## Spillover of gull-associated virus is uncommon

To compare which species were involved in host jumping, we reconstructed the phylodynamics of H13, H16 and HPAI H5 using a discrete traits model focused on taxonomic order. For H13 and H16, gulls acted as a source of virus to a limited number of other species, primarily wild ducks (H13: 42.0%; H16: 32.9% host jumps) and to a lesser degree shorebirds (H13: 20.7%; H16: 22.2%, Fig 3). Rarely did these two subtypes cross into domestic poultry, with the exception of a low number of H13 host jumps between gulls and gallinaceous poultry. Further, shorebirds (H13: 21.0%; H16: 23.1% host jumps) followed by ducks (H13: 10.3%; H16: 16.6% host jumps) were the only two hosts identified as sources responsible for transmission back to gulls (Fig 3). This was reflected in the high transition rates from wild ducks to gulls (H13: mean: 1.45, range: 0.49–2.41; H16: mean: 1.60, range: 0.48–2.72) and shorebirds to gulls (H13: mean: 1.38, range: 0.51–2.25; H16: mean: 2.36, range: 0.96–3.76). This pattern was remarkably

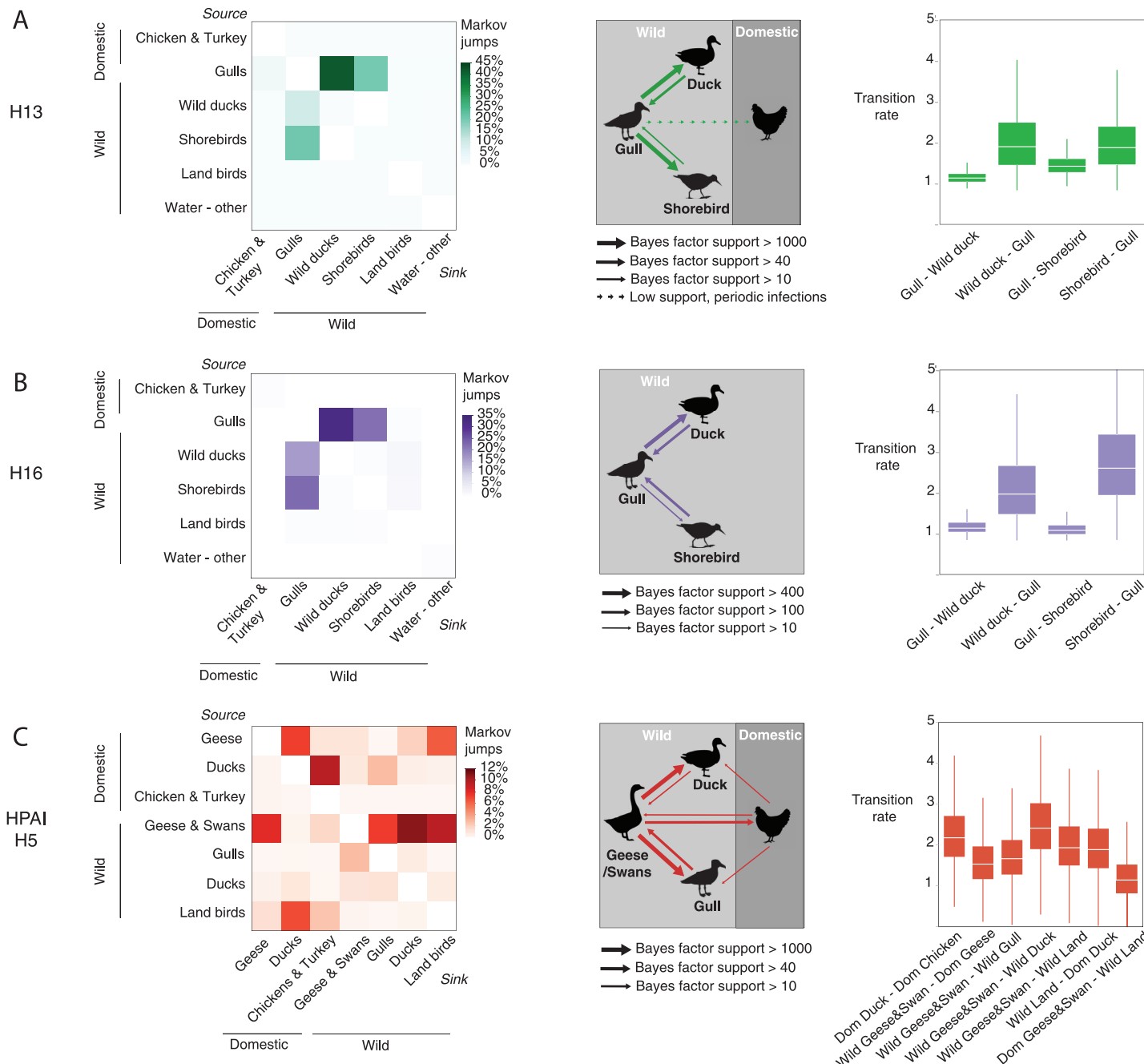

**Fig 3. Host jumps and spillover of influenza A viral subtypes at the wild-domestic interface reflects co-evolution with the reservoir host.** The heatmaps: A) H13 (green), B) H16 (purple) and C) HPAI H5 (red), indicate the frequency of transitions between host pairs estimated using a discrete trait phylogenetic model and the number of transitions between species/taxa was determined using Markov jumps. Host interfaces between wild and domestic birds (middle panel) are inferred based on host transitions that are well supported by Bayes Factors (BF). Dashed lines indicate known but infrequently sampled transmissions that could not be included in the analysis due to low sample size. Transition rates between host pairs are shown in the box and whisker plots (only the most strongly supported rates are shown). Large variation in transition rates is observed for HPAI H5N1 with highest rates between wild birds, followed by domestic poultry. In contrast, transition rates for H13 and H16 are lower with consistent patterns observed between host pairs for each subtype. Animal silhouettes from http://phylopic.org.

consistent for both the H13 and H16 subtypes and indicated a limited host range involving aquatic wild bird reservoirs.

## HPAI H5 spillover is multi-faceted involving more than ducks

For HPAI H5, a much wider range of avian hosts contributed to transmission dynamics, both domestic and wild. Our phylodynamic reconstruction largely focused on the evolutionary pathways of the most recent clades, including clade 2.2.1 and clade 2.3.4 and showed that spill-over is multi-faceted involving transmission chains other than just wild to domestic ducks. Within domestic species, geese played a key role as source of virus for domestic ducks (8.2% host jumps), that in turn served as a source of virus to chickens and turkeys (11.1% host jumps). Host jumps at the domestic-wild interface primarily occurred from domestic geese to land birds (6.0% host jumps, ie. corvids, passerines and raptors) and domestic ducks to gulls (2.4% host jumps). However, this type of domestic to wild spillover occurred at much lower frequency than transmission from wild to domestic hosts.

Host jumps from wild to domestic birds primarily involved dispersal from wild geese and swans to their domestic counterparts (7.8% host jumps). Additionally, land birds were an important source to HPAI H5 to domestic ducks (8.6% host jumps) and to a lesser degree galli-naceous poultry (3.1% host jumps). Within wild birds, geese and swans played an outsized role in the dispersal to wild ducks (12.4% host jumps), followed by land birds (9.9% host jumps) and gulls (8.5% host jumps). Of interest, gulls infrequently acted as a source of virus back to wild geese and swans (3.3% host jumps). Transition rates for HPAI H5 indicated that dispersal was most efficient between wild-to-wild hosts and domestic-to-domestic hosts, reflecting a barrier between the two host types. For example, the highest transition rates occurred between wild geese/swans to wild ducks (mean: 2.50, range: 1.66–3.34) and domestic ducks to chick-ens/turkeys (mean: 2.26, range: 1.51–3.01).

## Gulls contribute to rapid spread of HPAI and slow spread of LPAI

The rate of spread by different hosts was investigated for each subtype using a joint continuous geographic and discrete host diffusion model. On average, the HPAI H5 subtype spread at a rate of 4250 km/year among wild birds (95% HPD: 3009–5620) (Fig 4A), 4 times faster than H13 that spread at a rate of 1029 km/year (95% HPD: 360–1970) (Fig 4B) and over 8 times as fast as H16 that spread at a rate of 521 km/year (95% HPD: 281–773) (Fig 4C).

When diffusion rates were evaluated by each host taxonomic group, wild birds contributed to the fastest rates of geographic spread of HPAI H5 (Fig 4A), with gulls playing a key role in moving the virus more rapidly than any other host (5919 km/year, 95% HPD: 4057–7897). Wild ducks (4426 km/year, 95% HPD: 2848–6286) and geese and swans (4269 km/year, 95% HPD: 3478–5145) were also important for rapid spread, while land birds (2384 km/year, 95% HPD: 1653–3152) moved the virus much slower compared to aquatic species. The spread of HPAI H5 by domestic birds was relatively slow compared to wild aquatic species. Domestic geese contributed to fastest spread (2970 km/year, 95% HPD: 2346–3656), while ducks (1830 km/year, 95% HPD: 1326–2376) and chickens (1423 km/year, 95% HPD: 898–1977) were associated with the slowest rates of HPAI H5 dispersal.

For H13 and H16, which showed a more restricted host range, taxonomic groups were con-sidered at lower resolution to determine species-level contributions to spread. Our results showed substantial variation in the rates of spread between and within wild species. For H13, Glaucous-winged Gulls (*Larus glaucescens*) spread the virus fastest (1524 km/year, 95% HPD: 567–2603), followed by shorebirds (1224 km/year, 95% HPD: 407–2351), Black-headed Gulls (*Chroicocephalus ridibundus*) (1179 km/year, 95% HPD: 358–2161), Ring-billed Gulls (*Larus*

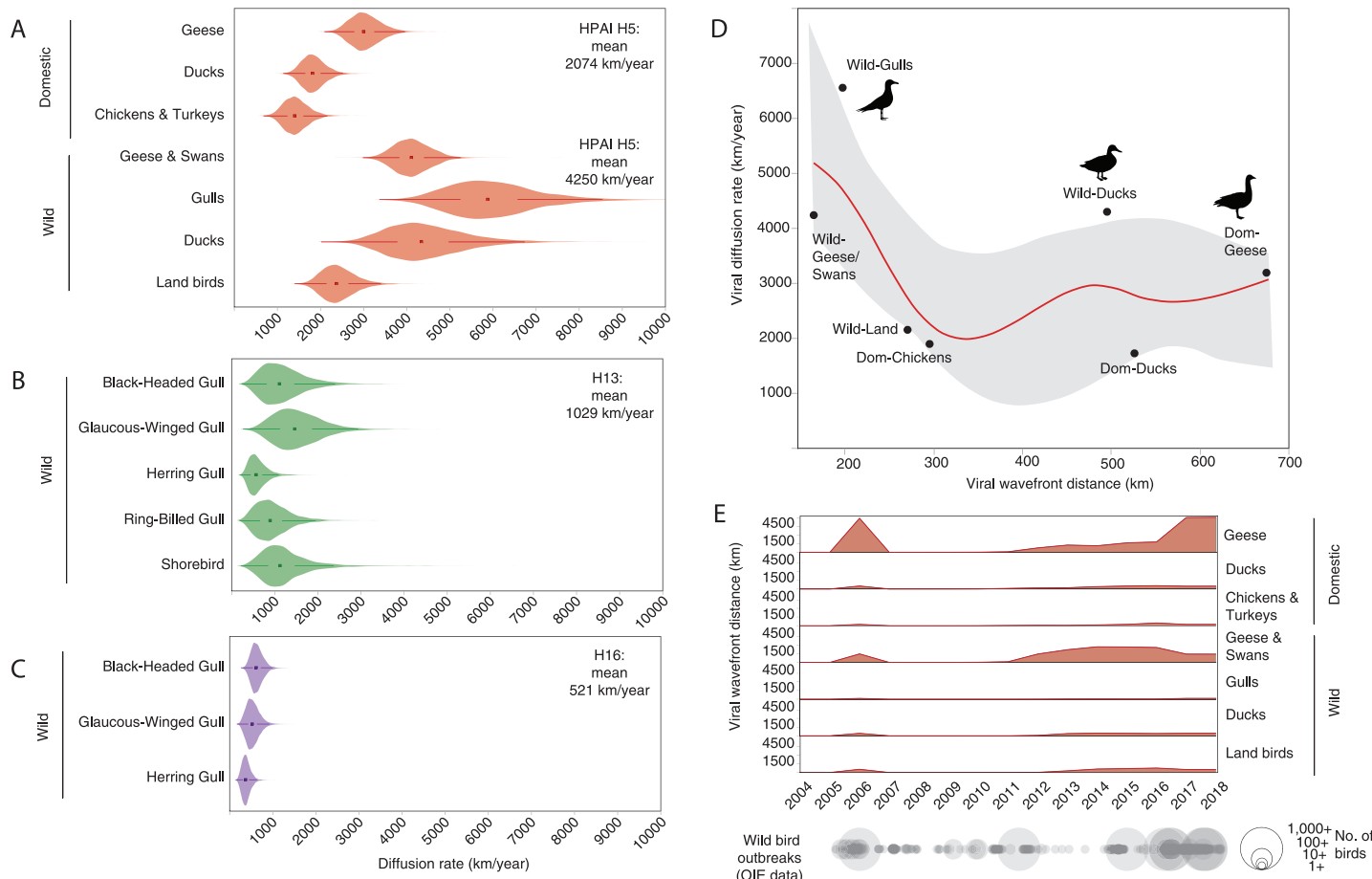

**Fig 4. Ecologically divergent hosts contribute to the diffusion and geographic expansion of influenza A virus.** A) The diffusion rates of highly pathogenic avian influenza (HPAI) H5 (red), B) H13 (green) and C) H16 (purple). D) Comparison of two dispersal metrics of HPAI H5: viral diffusion (km/year) and viral wavefront distance (km) plotted for each host type (round markers). The red line indicates the negative relationship between viral diffusion and wavefront distance of HPAI H5 based on regression analysis, with grey shading indicating the 95% confidence interval. E) Viral wavefront distances for HPAI H5 fluctuate over time with a large peak during 2005–2006 and a gradual increase from 2010 onwards. Dispersal rates (km/year) and wavefront distance (km) were estimated using a joint continuous geographic and discrete host diffusion model. Wild bird outbreak data are courtesy of the World Organisation for Animal Health. Available from: https://wahis.oie.int/#/home.

*delawarensis*) (951 km/year, 95% HPD 287–1752) and finally Herring Gulls (*Larus smithsonianus*) (605 km/year, 95% HPD: 262–1032) (Fig 4B). For H16, spread was much slower. Herring Gulls spread H16 the fastest (635 km/year, 95% HPD: 373–904), followed by Glaucous-winged Gulls (536 km/year, 95% HPD: 262–833) and Black-headed Gulls (385 km/year, 95% HPD: 202–581) (Fig 4C). We lacked samples from shorebirds to estimate dispersal rates for H16 and lacked samples from ducks to estimate dispersal rates for either H13 or H16.

## Domestic birds drive slow but steady range expansion of HPAI

To determine host-specific differences in the geographic expansion of the virus, the wavefront distance was estimated from the continuous geographic diffusion model. The wavefront distance tracks the distance from the origin of the epidemic to the location of currently circulating phylogenetic lineages, measured as a great circle distance and averaged across all lineages. For HPAI H5, an inverse relationship was suggested between wavefront distance and diffusion

rates (Fig 4D). Gulls (197.25 km) and wild geese and swans (164.68 km) contributed least to geographic expansion of the virus from detected origin. In contrast, domestic geese (676.93 km) and domestic ducks (526.04 km) were responsible for the greatest range expansion of HPAI H5. These trends were evaluated over time and indicated a 2005–2006 peak in expansion coinciding with an increasing frequency in wild bird outbreaks based on global case count data [29]. A second sustained expansion started in 2010 that coincided with the emergence of clade 2.3.4.4 based on molecular dating (95% HPD: 2009.21–2011.41). This period also coincided with an increasing frequency and magnitude of reported wild bird outbreaks across the globe (Fig 4E).

## Discussion

This study systematically compared the evolutionary dynamics of IAV subtypes across taxonomically distinct bird groups with the goal of informing how different birds contribute to global spread. We provide evidence that across avian hosts, the same subtype behaves differently in terms of global distribution, viral migration routes and spillover with differences observed at taxonomic scales much lower (ie. family and genus) than previously considered. Typically, global studies of IAV seek to resolve how birds contribute to transmission dynamics using broad categories such as 'wild' or 'domestic' or 'Anseriformes' or 'Galliformes', which may not be adequate for accurate prediction of IAV outbreaks. Surveillance of birds often reflects an uneven sampling strategy with regard to host taxa, often focused on wild and domestic ducks, or poultry, that are abundantly sampled and act as a source of highly pathogenic strains. Often hampered by sample size, especially for lesser studied groups such as gulls and shorebirds, the approach of aggregating bird species into larger taxonomic categories has been adopted out of necessity. This study sought to address this imbalance by characterizing influenza viruses from a diversity of wild bird species over a 10-year period, allowing for comparisons between host groups representative of the biodiversity in natural settings. Doing so provided a unique opportunity to clarify the mechanisms by which different bird hosts contribute to inter-hemispheric transmission and wild-domestic bird spillover.

Our findings provide empirical evidence that wild ducks occupy an optimal space for influenza that contributes to both spatial (wavefront) expansion and fast diffusion of HPAI H5. In contrast, gulls contribute to rapid diffusion but rarely spread the virus beyond its known range. Gulls were responsible for moving HPAI H5 more rapidly than any other host (5919 km/year), a finding that may reflect the long-distance, pelagic movements of gulls, relative to the more punctuated, land-based movements performed by wild ducks. This result broadly correlated with previous studies that investigated the higher taxonomic categories of Anatidae (ducks), Phasianidae (poultry) and Neoaves (non-waterfowl wild birds) that detected a large overall contribution of ducks to the epidemic wavefront of HPAI H5 [30]. Our findings also uncovered an inverse relationship between diffusion rates and wavefront expansion among both wild and domestic birds, implying a potential trade-off between the two. Rapid movement between source and sink locations may be efficient for spread, but may result in less spatial spread if infected birds interact with few susceptible hosts along the way, limiting the potential for onward transmission.

Wild geese and swans were identified as primary source hosts for HPAI H5, responsible for transmission leading to infection of wild ducks, gulls and land birds, as well as domestic geese, based on the results of the Markov jumps analysis. The uncovering of a complex HPAI H5 transmission network, driven largely by wild geese and swans, is a novel finding, suggesting Anserinae may be under-recognized as an important host relative to waterfowl. Experimental studies of Anserinae indicate geese and swans are capable of lengthy viral shedding, with a

mean of 5–6 days [31]. However, geese are characteristically more resilient as hosts, capable of asymptomatic shedding with the potential to cover longer distances while infectious. As such, the Bar-headed Goose (*Anser indicus*) has been implicated as a primary host for the Qinghai Lake outbreak seeded by populations migrating along the Central Asian flyway [32,33]. In contrast, swans are relatively more susceptible, show a higher proportion of deaths after infection and are viewed as sentinel species capable of limited localized transmission [31,34]. Factors such as pre-existing immunity, age, season and species are known to modify the susceptibility of Anserinae in nature [35,36], and may explain the large number of recent HPAI H5 clade 2.3.4.4 outbreaks involving both swans and geese across Eurasia and North America.

In terms of host abundance, geese are a prominent example of waterfowl that have successfully adapted to human activity with dramatically increasing populations during the late 20th century. Across the American continent, Snow Geese (*Chen caerulescens*), Greater White-fronted Geese (*Anser albifrons frontalis*) and Canada Geese (*Branta canadensis maxima*) have experienced exploding population numbers attributed to an increase in agricultural and suburban land use [37–39]. Similar trends have been observed for Greylag Geese (*Anser anser*) and Barnacle Geese (*Branta leucopsis*) that have reached super-abundance in Europe and Asia [40,41]. Attracted by crops and grass, the interface between herbivorous geese and livestock has expanded globally, with co-grazing having consequences for the transmission of infectious disease [42] that have yet to be adequately characterized for IAV. In this study, the most frequent route of HPAI H5 spillover involved wild geese and swans seeding infection in domestic geese, followed by onward transmission to domestic ducks. Investigation of HPAI H5 wave-front expansion over time supported that domestic geese have played the primary role in increasing the global range of the virus, far more than domestic ducks. Historically significant, domestic geese were the original hosts of the goose/Guangdong (Gs/GD) highly pathogenic strain of H5N1 that emerged in southeast Asia in 1996. Coupled with the global expansion of goose farming due to the versatility of products ie. meat, eggs, feathers and liver products [43] the interaction with wild geese may be creating a new wild-domestic interface that may rival free-grazing ducks in southeast Asia.

One of the limitations of this study was the dependence on publicly available sequences that are inherently biased with regard to host or geographic region. For example, active surveillance of wild birds is rarely performed in some African and Asian countries [25], resulting in less robust metrics for wild bird-mediated dispersal in these HPAI-endemic regions. We attempted to account for this in our downsampling strategy by including an equivalent number of samples for host taxa or geocluster when analyzing each subtype. However, the degree to which the subsequent analysis was still sensitive to uneven sampling effort remains unclear. This may have led to over-estimates of the role of domestic geese in the wavefront expansion of the HPAI H5 virus relative to wild birds. In addition, where wild bird surveillance is conducted there is a sampling bias towards sick or dead birds, or species that are more susceptible rather than resilient or asymptomatic birds. This highlights the scientific value in active surveillance of wild and domestic birds concurrently, rather than a focus on poultry in HPAI-endemic regions, as these interfaces represents a critical interface for viral spillover, and an opportunity to learn about the changing role of wild birds in HPAI epidemiology.

Beyond the narrative that gulls behave as a reservoir for the H13 and H16 subtypes, our results provided evidence of frequent bi-directional transmission to wild ducks and to a lesser degree, shorebirds. Gulls acted as the source of virus for ducks and shorebirds that rapidly transitioned back to the reservoir host, evidence that ducks and shorebirds are competent hosts, but their contribution to sustained transmission of H13 and H16 outside of gulls is limited. Similarly, the host range of H13 and H16 was restricted to water birds, with spillover to poultry (specifically turkeys) found to be rare. However, investigation of all three subtypes

indicated gull-poultry interactions do occur, as gulls acted as sink hosts for HPAI H5 in Europe and Asia after transmission from domestic ducks. The role of gulls as spillover hosts is complex and may be under-recognized in view of their numerical abundance worldwide. Gulls such as the Ring-billed Gull and Glaucous Gull (*Larus hyperboreus*) are among the most abundant birds in the world [44]. While spillover between gulls and poultry may be less common than other pairs of bird taxa, the ubiquity of gulls and their habituation to human resources, coupled with their capacity for rapid dispersal of HPAI suggests that monitoring of gulls at poultry and human interfaces would be useful for identifying the mechanisms of disease spread in the Anthropocene.

Investigation of gulls afforded the opportunity to examine co-circulation dynamics of H13 and H16 –two 'specialist' subtypes, that compete for the same host population. Prior models have proposed that the evolution of RNA viruses, including IAV, hinges on two dynamics: selective forces resulting from cross-immunity and non-selective epidemiological forces resulting from spatial structuring [45]. We found evidence that cross-protective immunity between lineages of H13 and H16 resulted in temporal structuring at the local-scale in the Cordova population of gulls. The two most common clades, H16 from Northern Europe and H13 from East Asia were strongly cross-reactive against H13 from South America, that infrequently occurred in the gulls. H13 from South America was the most broadly cross-reactive strain, which may be causal of the low levels of this clade circulating in the gull population (1 out of 10 years of the study). A limitation of the antigenic analysis is that the exposure history of the birds was unknown, therefore we cannot exclude that antibodies were elicited through infection from more than one subtype. We attempted to account for this by only including young birds (2–6 months) to maximize the likelihood that antibodies were *de novo* (not materially {Verhagen, 2020 #6039}derived) and generated against first IAV exposure. Future studies that aim to investigate cross-protective immunity in wild settings may benefit from longitudinal sampling of early life stages when birds have restricted mobility, or sampling of birds in human-modified environments where the possibility of recapture may be higher due to year-round supply of anthropogenic resources.

We also found evidence of a lack of cross-protective immunity between subtypes, for which the pattern of occurrence was best explained by spatial structuring by geographic origin. The two most common clades, H16 from Northern Europe and H13 from East Asia, were not cross-reactive. Antigenic analysis suggested that evolution in disparate geographic regions, rather than antigenic overlap was responsible for any fitness advantages that may explain turnover between the clades. Our results support that the mechanism of competition between and within subtypes is two-fold, driven by immune selection, as well as geographic isolation. This adds nuance to the view that antigenic selection acting on H13 and H16 is weak, relative to the pattern of geographic isolation and periodic inter-hemispheric transmission {Verhagen, 2020 #6039}.

Our analysis supports that the H5 subtype is antigenically unique and non-overlapping with any co-circulating lineages of H13 and H16 in Alaskan waterbirds. No gulls, and only one duck were positive for antibodies against LPAI H5 in the subset of birds tested, evidence that supports waterbirds inhabiting the Pacific Rim of North America being largely immuno-naïve against the H5 subtype [46]. This provides insights into how non-reservoir hosts that lack routine exposure to a subtype or clade, may actually serve to be effective at dispersal under specific conditions, as demonstrated by the introduction of HPAI H5 clade 2.3.4.4 via Alaskan wild birds [47–49] and the months-long spread across North American flyways [50]. In this study, gulls were linked with the fastest rates of HPAI H5 dispersal. Conversely, dispersal of H13 and H16 by gulls was substantially slower. The slower dispersal of gull-associated subtypes may reflect a higher level of population immunity that could reduce the number of individuals

involved in onward transmission. Adaptive immunity in gulls accumulates with each successive annual cycle, reaching 70% seropositivity in adult birds [51]. Introduction of novel strains coupled with the absence of broadly-neutralizing immunity, particularly in younger birds, may translate to conditions that favor rapid dispersal that are unusual in the reservoir host.

Lastly, we quantified the spatial distribution of IAV at the global scale and found clear evidence of subtype-specific dispersal dynamics. Few studies have explicitly tested whether global patterns differ or are consistent across HA subtypes, because of the focus on investigating a single HA subtype (typically H5 or H7) or alternatively, internal gene segments. Despite overlapping in geographic range, H13 and H16 showed different routes of entry between hemispheres and different regions where each subtype circulated extensively. Two geographic areas emerged as major sources of H13: the Mississippi-Atlantic in the US and the Black Sea-Mediterranean, each acting as a source of local-scale dispersal in temperate zones. In contrast, higher latitude regions were more important sources of H16, including Northern Europe and to a lesser degree, the Northern Atlantic, as well as Alaska. Underlying differences in thermostability due to climatic adaptations, antigenic interactions or both may explain the unique and complementary pattern of distribution observed between the two subtypes. Overall, the major thrust of inter-hemispheric movement across all three subtypes was both eastward via the Pacific Rim, and westward via the Atlantic Rim into North America. This concurs with studies of seabird and shorebird movement in the Pacific Rim that indicate a larger magnitude of eastward migration from Siberia into Alaska compared to the reverse direction [52]. We did not detect a prevailing signal of viral flow via the Pacific Rim into North America as focusing exclusively on the high-profile incursion of clade 2.3.4.4 HPAI H5 would imply. Evaluating multiple HA subtypes highlighted that gulls are capable of mediating long-distance dispersal via both the Pacific and Atlantic Rims, supporting proactive surveillance at high latitudes along the continental margins to achieve early detection of IAV.

The rapid rate of IAV evolution set against the backdrop of the Anthropocene underscores the need to perform global surveillance that is responsive to host-viral systems undergoing flux. Human activities create unique species assemblages that are not typical in the wild. Low biosecurity farms and live bird markets are considered an important interface where HPAI H5 expands its host range into land-based species that are atypical in wetlands where duck reservoirs occur. Our analysis identified the following major interfaces: (i) wild to domestic geese, (ii) synanthropic land birds to domestic ducks, and (iii) domestic geese to synanthropic land birds, as useful for early detection of IAV spillover between wild and domestic systems. Taxonomic relatedness of host species alone could not explain the involvement of land birds in spillover transmission, hinting at the potential for numerical abundance of 'pest' birds such as sparrows, pigeons, starlings or crows to act as an important ecological driver. Our study showed that HPAI transmission is complex and multi-faceted involving numerous transmission chains other than just wild to domestic waterfowl that have traditionally been a focus of surveillance.

## Methods

### Ethics statement

Procedures for capture and handling of wild birds were approved by the Tufts University Institutional Animal Care and Use Committee (Protocol # G2017-118) and the U.S. Geological Survey Bird Banding Laboratory (Federal Bird Banding Permit: 21963 and 23432).

### Sample collection

Wild birds representing the orders Anseriformes and Charadriiformes were sampled in Alaska and Massachusetts in North America between 2008–2018. In Alaska, wild waterfowl primarily

dabbling ducks including Mallard (*Anas platyrhynchos*), Northern Pintail (*Anas acuta*) and American Green-winged Teal (*Anas carolinensis*) were sampled from 2008–2012 at Minto Flats State Game Refuge, interior Alaska (lat. 64.90, long. -148.85), a wetland complex that serves as a major breeding, molting and staging area. Birds were captured and swabbed for oral-pharyngeal and cloacal samples, and blood was collected from the jugular or cutaneous ulnar vein in the wing. Detailed methods for capture and influenza sampling are published in Hill, Ma [53].

Gulls and shorebirds were sampled at Cordova, southcentral Alaska (lat. 60.59, long. -145.77) between 2009 and 2018, a breeding and stopover site for wild birds migrating along the Pacific Rim. Glaucous-winged Gulls were captured with a net launcher during the breeding and post-breeding season (April-September), while Least Sandpipers (*Calidris minutilla*) and Western Sandpipers (*Calidris mauri*) were captured with mist nets during the spring migration (April-May). Oral-pharyngeal and cloacal swabbing was performed, and blood was collected from the brachial vein. Fecal sampling was performed weekly at congregation sites for gulls and shorebirds from March through October.

On the Atlantic Rim, gulls were captured throughout Massachusetts (lat. 42.36, long. -71.06) at urban habitats that attract migratory and over-wintering gulls. Herring Gulls (*L. smithsonianus*), Ring-billed Gulls (*L. delawarensis*) and Greater Black-backed Gulls (*L. marinus*) were captured with a net launcher and birds were sampled as per Ineson, Hill [51]. Sampling was performed weekly from October to April coinciding with the non-breeding season between 2012 and 2014. All swab samples were stored in VTM (Remel) at 4˚C in the field for up to 12 hours and transferred to -20˚C or -80˚C storage prior to laboratory analysis.

## Influenza screening

Viral RNA was extracted from swab samples using the Mag-Bind Viral DNA/RNA 96 Kit (Omega Bio-Tek Inc., Norcross, Georgia, USA) on a semi-automated KingFisher Purification System robot (ThermoFisher Scientific, Waltham, Massachusetts, USA). One-step real-time RT-PCR was used to detect influenza RNA using primers targeting the matrix gene [54] and qScript XLT 1-Step RT-qPCR ToughMix (Quanta BioSciences, Gaithersburg, Maryland, USA). Samples with a cycle threshold (Ct) value less than 45 were considered positive.

To amplify virus, VTM (100 μl) from positive samples was inoculated into the allantoic cavity of 9- to 11-day old embryonating specific pathogen-free chicken eggs (Charles River Laboratories, Wilmington, Massachusetts, USA) and incubated at 37˚C for 72 hours or until embryo death, as detected by daily candling. RNA was extracted from the allantoic fluid and matrix rRT-PCR was performed to screen for presence of IAV [55]. The eight genomic RNA segments of IAV were simultaneously amplified using a multi-segment RT-PCR strategy [56]. Illumina libraries were prepared using the Nextera DNA sample preparation kit (Illumina, Inc., San Diego, California, USA) with half-reaction volumes as described previously [57]. Whole-genome sequencing was performed on the DNA amplicons at the J. Craig Venter Institute in Rockville, Maryland, USA as described by Nelson et al. [58] and all sequences deposited into GenBank (S1 and S2 Data). Subtyping was confirmed by BLASTn of the sequence against isolates in GenBank and identifying the subtype match that showed highest percentage identity.

## Antigenic characterization of viruses

To test for the presence of influenza antibodies in serum, an Enzyme-Linked Immunosorbent Assay (ELISA) kit (IDEXX AI MultiS-Screen, Westbrook, Maine, USA) was used. Absorbance values were measured with an Epoch Spectrophotometer (BioTek Instruments, Winooski,

Vermont, USA) and the mean absorbance value for each sample was used to calculate the sample to negative ratio. Samples with a ratio ≤0.5 were considered positive for influenza antibodies. Seropositive samples from Alaskan gulls and ducks were selected to assess cross-reactivity between strains and subtypes using hemagglutination inhibition (HI) assays as per the methods of Puryear, Keogh [59]. Only sera from flighted juveniles (between 2 and 6 months) were assayed in order to maximize the likelihood that antibodies were *de novo* (not materially derived) and were generated against first IAV exposure. Serum was defined as positive for HI titers equal to 1:32 or above.

The antigen panel for the HI assay included representatives of H5, H13 and H16 collected from Alaskan gulls and ducks belonging to different geographic origins inferred by phylogenetic analysis of the HA segment. We estimated the geographic source of the most recent common ancestor (MRCA) of selected antigens using ancestral state reconstruction applied to each HA phylogenetic tree. All nodes that were used to infer the geographic origin of the MRCA were statistically supported by a posterior probability >0.95. The MRCA was identified from one of the prominent geographic clusters ('geoclusters') from across the globe, and implemented using BEAST v1.10.4 [60,61] following the methods described in detail below.

The HI assay included two viral antigens for each of the three HA subtypes, a total of six antigens. For H5: A/sandpiper/Southcentral Alaska/16MB01156/2016(H5N2) and A/sandpiper/Southcentral Alaska/16MB01220/2016(H5N2) both originating from Pacific-Central United States (US) inferred from the MRCA. For H13: A/Glaucous-winged Gull/Southcentral Alaska/17MB03606/2017(H13N2) originating from East Asia and A/Glaucous-winged Gull/Southcentral Alaska/ 15MB01632/2015(H13N6) originating from South America. For H16: A/Glaucous-winged Gull/Southcentral Alaska/16MB02941/2016(H16N3) originating from Northern Europe, and A/Glaucous-Winged Gull/Southcentral Alaska/20MB02834/2020 (H16N3) originating from Mississippi-Atlantic U.S.

To assess the degree of antigenic overlap between viruses, the titers from the HI assay were visualised using a multiple correspondence analysis (MCA) performed using JMP Pro v.14.3 [62]. The MCA uses categorical data as input, which for this study included i) the geographic origin associated with each virus based on MRCA analysis, and ii) the reactivity of the sera against each virus (HI present/absent). Through representation of these variables in two-dimensional Euclidean space, clustering of viruses was assessed and interpreted as antigenic overlap. In contrast, viruses that occupied different spaces in the MCA plot were interpreted as antigenically distant.

## Linking viral sequences, geography and host

A total of 888 complete influenza sequences representing 12 HA subtypes were generated from our North American wild bird surveillance between 2008 and 2018. This included H13 (n = 23 sequences) and H16 (n = 58 sequences) used in this study. For comparative analysis, all HA sequences belonging to HPAI H5, H13 and H16 subtypes were obtained from the Global Initiative on Sharing All Influenza Data (GISAID) on 20[th] September 2019. Search criteria included complete sequences from avian hosts for which the collection year and month was known, as well as geographic location.

For each geographic location we extracted centroid coordinates for the lowest administrative unit using Geonames.org. All sampling coordinates were input into a k-means cluster analysis to determine groupings of sampling sites based on geographic proximity rather than administrative or political boundaries (implemented using JMP Pro v.14.3 [62]). A total of 11 geoclusters were identified that corresponded to Alaska, Pacific-Central US, Mississippi-Atlantic US, Northern Atlantic, South America, Northern Europe, Central Europe, Africa, Black

Sea-Mediterranean, Central Asia, and East Asia (S4 Text). These geographic 'traits' were used for subsequent phylodynamic analysis.

Hosts were grouped by two traits: species and family/sub-family according to eBird.org (a recognized global authority on ornithological taxonomy). Any environmental samples were removed from the dataset. Host taxa were further parsed as 'wild' or 'domestic' and any ambiguous data (ie. "duck", "goose" or "avian") were removed from the dataset. Ducks or geese were only included if the metadata specified whether the sample was from a wild or domestic bird, or the full species names was provided, allowing inference based on history of domestication in a region (ie. swan-goose in China was considered 'domestic', pink-footed goose in the Netherlands was considered 'wild').

## Downsampling global datasets

To mitigate sampling bias, datasets for each subtype were stratified by geocluster and downsampled to ensure an equivalent number of samples across the 11 geographic traits. Samples were limited to 2005–2018 but were not stratified and downsampled by year because the spatio-temporal distribution was considered an important feature linked to the epidemiology of each subtype. Alignments for each subtype were performed using default settings in MUSCLE v.3.8.31 [63] and trimmed to the open reading frame. For each geocluster (ie. Africa or East Asia), maximum likelihood phylogenetic trees were constructed using RAxML v8.2.12 [64] using the GTRGAMMA substitution model. A general time reversible model was chosen to account for the complexity of the evolutionary process that governs how influenza viruses mutate. Each tree was downsampled to approximately 40 taxa while preserving the maximum amount of genetic diversity using the Phylogenetic Diversity Analyzer tool (www.cibiv.at/software/pda) following the approach of Trovao, Suchard [30]. This downsampling approach was repeated for each subtype stratified by host traits (S5 Text). Each of the subtype datasets was analyzed separately. Analyses were performed to assess spatial spread according to geoclusters: H13: n = 398, H16: n = 191, and HPAI H5: n = 295. Separate analyses were performed to assess spread according to host taxa: H13: n = 298, H16: n = 202, and HPAI H5: n = 367.

## Bayesian phylogeny reconstruction

All phylogenetic tree reconstructions using Markov chain Monte Carlo (MCMC) sampling analyses were performed using BEAST v1.10.4 package [60,61] in combination with the BEAGLE library [65]. The dataset included viral sequences collected between 1972 and 2019 to ensure accurate rooting for the tree topology, but sequences prior to 2005 were masked in the phylogeographic analysis as per Hicks, Dimitrov [66]. This allowed us to analyze IAV evolution over a more recent time period that also represented the bulk of available data. Phylogenies were reconstructed using a General Time Reversible nucleotide substitution model with gamma distribution of substitution rates, a Gaussian Markov Random Field Skyride coalescent model and an uncorrelated lognormal clock. Four independent MCMC runs of 200 million chains were performed for each segment. Runs were combined to ensure an effective sample size (ESS) over 200 was achieved in Tracer v1.6.0 and the maximum clade credibility (MCC) tree was determined. A subset of the last 500 trees from the posterior distribution was used to generate an empirical tree distribution used in the subsequent phylogeographic analysis, an approach that reduced computational time and burden [30,67].

## Phylodynamics incorporating geography and host

To estimate the spread of virus between geoclusters, a joint discrete trait and continuous geographic diffusion model was performed [30]. For discrete traits, the asymmetric substitution

model was selected to estimate pairwise bi-directional viral flow between geoclusters. A Bayesian stochastic search variable selection (BSSVS) was used to summarize the diffusion rates and Bayes factors (BF) [68] were estimated using SPREAD v1.0 [69]. Rates were considered statistically supported when BF >3.0, strongly supported when BF > 10 and decisively supported when BF >100 [70].

To estimate the Markov jump counts between geographic states along the nodes of the trees we applied the continuous-time Markov chains (CTMCs) model [71]. The number of jumps between states was expressed as a proportion of the total number of transitions occurring across the tree and depicted by heatmaps. Heatmaps were plotted using JMP Pro v.14.3 (SAS Institute Inc, Cary, North Carolina, USA). To quantify the proportion of time spent by the virus in each of these geographic states, Markov rewards were also estimated [71].

To complement the discrete traits analysis, we estimated the spread of virus through space using the continuous traits (latitude and longitude) of sampled taxa. To account for variation in the diffusion rates along branches, a Cauchy relaxed random walk model of evolution was selected [72] with 0.001 random jitter added to spatial coordinates to reduce overlapping. The reconstructed dispersal history in continuous space was visualised using SPREAD v1.0 and qualitatively assessed to understand the degree of inter-hemispheric transmission among the H5, H13 and H16 subtypes, and specifically trans-Atlantic and trans-Pacific Ocean movements of the virus.

To elucidate the role of different avian hosts in the movement of viruses, we quantified host-specific diffusion rates and geographic expansion [30]. The host-specific diffusion rate (km/year) is the mean speed the virus travels within a host group, and represents a collective estimate made up of multiple individuals and species. The host-specific geographic expansion or 'wavefront distance' (km) measures the distance from the origin of the epidemic to the location of currently circulating lineages per host group, measured as a great circle distance and averaged across all lineages [30]. Both measures were estimated from the joint discrete (host) trait and continuous (geographic) diffusion model, performed independently for each of the three subtypes. Although both discrete and continuous process are modelled separately, the joint inference allows host-specific contributions to the spatial dispersal dynamics to be estimated [30]. To account for transmissions among multiple hosts along the length of the branch, the latitude and longitude at each time point is used to estimate statistics from the compound host-specific trajectories.

To evaluate viral flow between different avian host taxa or 'host jumping' a discrete trait phylodynamics analysis was performed. For HPAI H5, host taxa were considered at the higher taxonomic resolution of family or sub-family and parsed into 'wild' or 'domestic' categories (ie. domestic geese). For H13 and H16, subtypes that have a narrower host range, host taxa were considered at both the sub-family and species level. To assess BF support between pairwise host taxa, BSSVS analysis was performed as per geographic discrete traits. Markov jumps counts between host taxa along the nodes of the tree were estimated using the CTMC model. To estimate the contribution of host taxa to the dispersal and geographic range expansion of the three subtypes, we performed a joint analysis of discrete and continuous spatial diffusion. A joint analysis allows the delineation of host-specific rates of spread from the continuous spatial dynamics model as per Trovao, Suchard [30].

## Estimates of viral interactions at the local scale

A separate local-scale analysis of H13 and H16 was performed, that included 134 viral sequences exclusively from our study site in Cordova, Alaska collected between 2009 and 2018. Phylogenetic trees of H13, H16 and PB2 were estimated using identical methods as described

above. Changes in relative genetic diversity for the two HA subtypes were estimated using the uncorrelated lognormal relaxed-clock with Gaussian Markov random field smoothing of the Bayesian skyride plot [73]. These plots showed changes in the viral effective population size (Ne) over time and were visualised in Tracer v.1.6.0. and plotted using JMP Pro v.14.3 [62].

To evaluate the degree of overlap and interaction between H13 and H16 within Cordova, we estimated rates of reassortment by quantifying transitions of the PB2 internal segment between subtypes. This was modelled using a a discrete trait phylodynamic analysis, in which the traits were represented as different HA subtypes. To evaluate BF support between pairwise HA subtypes, BSSVS analysis was performed (as per discrete traits analyses described above) and results were visualized as heatmaps.

## Supporting information

**S1 Text. Phylodynamics of H13 (A), H16 (B) and highly pathogenic H5 (C) subtypes based on the hemagglutinin gene time-calibrated phylogeny.** The left panels indicate the branching pattern and estimated divergence times of lineages (blue bars: 95% highest posterior density). The right panels indicate geographic origin of viruses (color-coded branches) estimated using a phylogeographic discrete trait model with Bayesian inference (Beast v10.4).
(DOCX)

**S2 Text. Temporal pattern of H13 and H16 circulation in gulls over a 10-year period in Cordova, Alaska.** A & B) H16 emerged from two geographically-distinct clades (blue and purple) that predominate, with introductions of H13 occurring periodically from East Asia (green) and South America (yellow). Geographic origins were determined by most recent common ancestor (MRCA) phylodynamic analysis. B) The temporal distribution of H13 and H16 virus lineages mapped onto two-dimensional space. The temporal distribution of the strains and subtypes in Cordova, Alaska indicates that the two clades of H16 from Northern Europe and the Mississippi-Atlantic US rarely co-circulate, suggestive of geographic structuring and high antigenic cross-reactivity. In contrast, the temporal overlap between H16 (Mississippi-Atlantic clade) and H13 (East Asian clades) is indicative of low competition or antigenic cross-reactivity.
(DOCX)

**S3 Text. Transitions of the PB2 segment between HA subtypes circulating within the Cordova gull population in Alaska (2009–2018).** These estimates quantified reassortment and were based on a discrete trait model. As indicated by high Bayes factor support, H16 was a major source of the internal segment to H13 and to a lesser degree H13 acted as a source of PB2 to H16.
(DOCX)

**S4 Text. Spatial clustering method for influenza A virus sequences used to define 'geoclusters'.** For each viral sequence of H13, H16, and H5 (low and highly pathogenic) we extracted centroid coordinates for the lowest administrative unit using Geonames.org. All sampling coordinates were input into a k-means cluster analysis to determine groupings of sampling sites based on geographic proximity rather than administrative or political boundaries (implemented using JMP Pro, SAS). A total of 11 unique geoclusters were identified across subtypes that corresponded to geographic 'traits' used for subsequent phylodynamic analysis. The base map is made with Natural Earth (naturalearthdata.com).
(DOCX)

**S5 Text. Downsampling method for influenza A virus sequences.** Datasets were stratified by location attributes (geocluster) and downsampled to ensure an equivalent number of samples for each the H13, H16 and HPAI H5 subtypes. For each geocluster (ie. Africa or East Asia), maximum likelihood phylogenetic trees were constructed using RAxML v8.2.12 (Stamatakis 2006) using the GTRGAMMA substitution model. Each tree was downsampled to approximately 40 taxa while preserving the maximum amount of genetic diversity using the Phylogenetic Diversity Analyzer tool ([www.cibiv.at/software/pda](www.cibiv.at/software/pda)) following the approach of Trovao *et al.* (2015). This down-sampling approach was repeated for each subtype stratified by host traits.
(DOCX)

**S1 Data. Accession numbers for influenza A viral sequences of the H13 subtype generated by this study.**
(TXT)

**S2 Data. Accession numbers for influenza A viral sequences of the H16 subtype generated by this study.**
(TXT)

# Acknowledgments

This global-scale study would not be possible without important wild bird samples collected by many dedicated people, including Andrew Ramey, Andrew Reeves, David Stallknecht, Pedro Jimenez-Bluhm, Nicola Lewis, Josanne Verhagen and Ron Fouchier. We wish to thank Brandt Meixell, Jillian Whitney, Yianni Laskaris, Kirsti Jurica and Ann Harding for valuable assistance during field work. The content is solely the responsibility of the authors and does not necessarily represent the official views of the National Institutes of Health, the U.S. Fish and Wildlife Service or the U.S. Geological Survey. Any use of trade, firm, or product names is for descriptive purposes only and does not imply endorsement by the U.S. Government.

# Author Contributions

**Conceptualization:** Nichola J. Hill, Mary Anne Bishop, Jonathan A. Runstadler.

**Formal analysis:** Nichola J. Hill, Nídia S. Trovão, Jonathon D. Gass, Jr.

**Funding acquisition:** Nichola J. Hill, Mary Anne Bishop, Jeffrey S. Hall, Jonathan A. Runstadler.

**Investigation:** Nichola J. Hill, Mary Anne Bishop, Katherine M. Ineson, Anne L. Schaefer, Wendy B. Puryear, Katherine Zhou, Alexa D. Foss, Daniel E. Clark, Kenneth G. MacKenzie, Jonathon D. Gass, Jr., Laura K. Borkenhagen, Jeffrey S. Hall, Jonathan A. Runstadler.

**Methodology:** Nichola J. Hill, Mary Anne Bishop, Nídia S. Trovão, Katherine M. Ineson, Anne L. Schaefer, Wendy B. Puryear, Katherine Zhou, Daniel E. Clark, Kenneth G. MacKenzie, Jeffrey S. Hall, Jonathan A. Runstadler.

**Project administration:** Wendy B. Puryear, Alexa D. Foss.

**Writing – original draft:** Nichola J. Hill.

**Writing – review & editing:** Mary Anne Bishop, Nídia S. Trovão, Jonathon D. Gass, Jr., Jonathan A. Runstadler.

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
