## [Decision Letter · Decision Letter 0]

16 Feb 2022

Dear Dr Hill,

Thank you very much for submitting your manuscript "Ecological divergence of wild birds drives avian influenza spillover and global spread" for consideration at PLOS Pathogens. As with all papers reviewed by the journal, your manuscript was reviewed by members of the editorial board and by several independent reviewers. The reviewers appreciated the attention to an important topic. Based on the reviews, we are likely to accept this manuscript for publication, providing that you modify the manuscript according to the review recommendations.

The paper has been evaluated by three reviewers and they have raised various issues about the manuscript. The points made by the reviewers are related -but not limited- to explicit discussion about limitations of the data obtained due to sampling bias, supporting (or toning down) some claims, as well as issues raised about the interpretation of serological data. Addressing all the points raised by the reviewers is unlikely to require significant experimental work, thus I recommend minor revisions. Once the authors address the comments made by the reviewers I will be able to (hopefully) make a final recommendation.

Sincerely,

Pablo Ramiro Murcia

Guest Editor

PLOS Pathogens

Marco Vignuzzi

Section Editor

PLOS Pathogens

Kasturi Haldar

Editor-in-Chief

PLOS Pathogens

orcid.org/0000-0001-5065-158X

Michael Malim

Editor-in-Chief

PLOS Pathogens

orcid.org/0000-0002-7699-2064

The paper has been evaluated by three reviewers and they have raised various issues about the manuscript. The points made by the reviewers are related -but not limited- to explicit discussion about limitations of the data obtained due to sampling bias, supporting (or toning down) some claims, as well as issues raised about the interpretation of serological data. Addressing all the points raised by the reviewers is unlikely to require significant experimental work, thus I recommend minor revisions. Once the authors address the comments made by the reviewers I will be able to (hopefully) make a final recommendation.

Reviewer Comments (if any, and for reference):

Reviewer's Responses to Questions

**Part I - Summary**

Reviewer #1: Hill et al present an exciting effort to partition the epidemiological and evolutionary roles of different host species which form the reservoir of an important zoonosis. The analysis nicely integrates diverse lines of molecular, serological and spatiotemporal evidence and is carefully carried out. Their comparative approach of analyzing different viral subtypes and host species is rare in the literature but is shown here as a powerful way to differentiate how different hosts affect viral spread and how species interact, ecologically and through cross-immunity. The manuscript is well written and the figures nicely illustrate the main messages. There are a handful of statements which I found slightly speculative and should be addressed either with further analysis or caveats to the text. The authors could also further go a bit further to convince readers that it is possible to confidently link dynamics to particular host species/taxonomic groups given the levels of cross-species transmission observed. I nevertheless support publication, pending these and some other relatively minor revisions.

Reviewer #2: The paper entitled “Ecological divergence of wild birds drives avian influenza spillover and global spread” By Hill et al. provides important insights on the role of different birds on the dispersal of avian influenza viruses, focusing on gull-associated subtypes, H13 and H16, and a more generalist subtype, HPAI H5. Differently from previous works, they have investigated the role of avian hosts in the spatial expansion and speed of diffusion of the virus at lower taxonomic scales, providing evidence of the contribution of gulls, wild ducks, land birds, wild geese and different domestic birds in the virus diffusion dynamics. Moreover, they performed in depth evolutionary analyses of the three subtypes over an evolution period of 10 years. Analyses are well conducted and the methods clearly described.

My main concern regards the possible bias in the sequence data available, which, in my opinion, has not been sufficiently highlighted in the discussion. In particular, for the HPAI H5 it is important to consider that surveillance activity in wild birds is never or rarely performed in some countries (i.e. some African and Asian countries). In such geographic areas, often only sequences of viruses collected from poultry are available. I wonder if this may have affected the results of the analyses performed in this study. In particular, the investigation of the HPAI H5 wavefront expansion suggests that domestic geese have played a primary role in increasing the global range of the virus (lines 423-425). I wonder if this result could have been affected by the lack of sequences from wild birds from some areas. Moreover, considering that several studies (including this one) indicate wild birds as the major cause of virus spread and introduction in new territories, as well as the major source of the virus for poultry, I was trying to figure out if it is correct to include viruses from poultry in the wavefront analysis.

Another possible bias the authors should consider is that in most cases surveillance in wild birds is only passive. The authors should take into account that HPAI H5 infections may have different outcomes in different wild bird species and there may be a sampling bias towards the more susceptible species (where the infection is associated to a high mortality rate).

All these aspects should be considered and better discussed.

Reviewer #3: This is a nice study that contrasts the spatio-temporal dynamics of H13 and H15 subtype influenza viruses that predominantly circulate in gulls with the generalist highly pathogenic H5 viruses to infer the contributions of different host species to virus spread.

The study includes new sequences and serological data from US, and is analysed comprehensively along with global GISAID data using established Bayesian phylogenetic methods. The dynamics is presented with a greater taxonomic resolution than previously considered and I agree with the authors the analysis shed new insights that have potential for refining global surveillance. The manuscript is mostly well written, however a major weakness of the study is that the limitations associated with sequence data (gaps in surveillance), as well as the antigenic analysis is not discussed adequately.

**Part II – Major Issues: Key Experiments Required for Acceptance**

Reviewer #1: None

Reviewer #2: (No Response)

Reviewer #3: 1. The authors analyse serum from Alaskan gulls and ducks with unknown infection history to assess reactivity to a panel of H13 and H16 antigens from different clades. When a serum reacted with viruses from different subtypes the authors assume this as cross-reactivity between the strains. However, can the authors exclude that the serum contents of a bird were elicited through infection from only one subtype ? Suggest to address the limitations to this analysis in the Discussion.

2. Uneven sampling has been addressed nicely through downsampling (Fig S3), however it is not clear if such such downsampled data is adequate for robust estimation of H13 and H16 phylogeography over 10 years? While there is a lack of methods for testing data adequacy for geographic analysis, there are studies that have highlighted these limitations, which could be highlighted in the Discussion.

**Part III – Minor Issues: Editorial and Data Presentation Modifications**

Reviewer #1: One recurring issue is cases where patterns in the data are visually suggestive, but no formal analysis is presented to link the observed data to the broader biological explanation presented. Examples are results related to proximity to east Asia (~L200), the section on competitive exclusion (~L226), and the negative correlation (or regression?) between wavefront distance and diffusion rate for which statistics are alluded to, but not shown. Please tone down the language in these sections and in the discussion to be consistent with the speculative nature of the evidence, provide more robust analyses, or make verbal arguments for why no alternative explanations are possible.

L43. Statement on extending models of viral evolution is unclear. It would be nice to explain in what way these models were extended. This isn’t clear in the later text either.

L103. If gulls are already known to have a key role in inter-hemispheric spread of IAVs, what is new about the new question being posed here? Is this statement specific to H13 and H16? Similarly in next sentence, explaining what exactly the ‘pressing need’ is (i.e., what would better understanding of IAV ecology enable which is not currently possible?) would convey significance more clearly.

L145. A brief summary of the dataset/study design at the start of the results or towards the end of the introduction would be helpful to understand the results section. (I gave up half way through the results and skipped to the methods). For example, number of viruses sequenced per subtype and per bird species; whether the timescale and geographic provenance of samples was comparable across subtypes; and how the samples were collected (active/passive sampling?).

L153. The difference between H13 and H16 in terms of multiple clade circulation is difficult to see in Fig S5. It appears that representatives of all 3 clades of H16 were found in the most recent year, indicating sustained transmission. For H13, clade 2 is only found in Continental Europe and clade 1 seems to have gone extinct, which is inconsistent with statements in the text related to continued detection of all 3 clades. Please clarify how these conclusions were reached or revise the text.

L162. ‘same avian reservoir’ is misleading. Surely the different subtypes infect bird species at different rates. It seems plausible that these infections of birds with different host life histories influence viral evolutionary rate. Similar issue on L174 – you are dealing with a community of hosts, not multiple subtypes adapted to the same host species, or is cross-species transmission so rare that it can be ignored? I suggest a clarification here.

L21. Define ‘rare’, for example as the proportion of supported dispersal events between NA and Eurasia for each lineage.

L245. It would be interesting to overlay the dynamics of viral effective population size for the two H16 clades separately in Fig 2C to see if they mirror the expected epidemiological dynamics, but I understand this is considerable work and not vital to the conclusions of the study, so fine with whatever the authors decide.

L268. Perhaps a discussion point, but isn’t it possible that the lack of introductions from South America to Alaska reflects bird migratory patterns more than cross-reactivity? This seems more consistent with the data given that South American H13 viruses have no problem dispersing to other areas of North America (Fig. 1) where presumably there are other H13 and H16 viruses with which it might cross-react. A broader statistical analysis of pairwise cross reactivity vs spatiotemporal overlap (e.g., proportion of years where each pair of strains overlapped in each location where both was reported at least once) might be a better way to formally test if cross reactivity is affecting viral invasion success, but even that would just be a correlation.

L282. By “host barriers” do you mean physiological or ecological (ie, ducks and gulls don’t commonly live/forage in the same habitats)? Please clarify.

L290. Provide the denominator for the percentages of host jumps.

L516. Provide sample sizes for birds caught, samples tested etc.

L555. Presumably WGS was done on the amplified virus not field collected samples, but be explicit about this.

L600. Numbers don’t add up to 888 sequences unless you had 807 H5 sequences. Please clarify.

L625. Sentence on 2005-2018 sequences not being downsampled is unclear. Are you saying that tree tips, not their underlying sequences were downsampled or that only sequences from certain years were excluded from downsampling?

L679. Does the host-specific geographic expansion somehow account for the possibility that multiple species/host groups could have contributed to spread? If, not independent spread from the MRCA to the current range limit by each host group seems unrealistic. The lines below suggests that latitude and longitude data are incorporated and I see how that might work for a host specific diffusion rate, but please elaborate if/how it corrects for the issue specifically for the wavefront distance metric.

Fig 1. Need to define the lines and shading in the right column (maps)

Reviewer #2: Line 153. Figures should be numbered according to the order of appearance in the text.

Line 160. Replace “result” with “results”

Lines 163-165. Please, provide a brief explanation in the introduction of the classification system used for the HPAI H5 of the Gs/GD lineage.

Line 164. Clade 2.3.4.4 expanded not only into continental Europe, but also in Africa and several Asian countries.

Line 168. The range “November 2014-July 2015” is not correct (2004-2005?). Please modify.

Lines 635-637. It is not clear if the datasets were analysed separately and for which analysis each dataset was used. I suggest naming the dataset and mentioning for each analysis the dataset used.

Reviewer #3: 1. The antigenic panel in Figure 2 is named with location of origin. For consistency, consider include the clade numbers shown in Fig S5.

2. The language surrounding MRCA/origins can be cleaned up a little throughout the ms. For example, “H16 consisted of two geographically-distinct clades (Northern Europe and Mississippi-Atlantic US).” should be “originated/emerged from two …”

3. Lines 238-240. “but rapidly accrued with the introduction of the South American clade, as inferred from Bayesian skyride plots that showed changes in viral effective population size over time.” is not apparent from Figure 2B. Instead, there appears to b a gradual increase in relative genetic diversity until 2015, followed by fluctuations.

4. Suggest to rephrase Line 81 “as relative extremes on the (avian) phylogeny” is not clear.

5. In Fig 2 legend clarify the dashed lines used in Fig2 F and G.

6. It would be useful it the clade numbers used in Fig S5 were incorporated in the description of the viruses, rather than just MRCA.

8. Avoid “very in Discussion Line 369, or describe the magnitude of differences.

9. Minor typo “out (of) necessity”

PLOS authors have the option to publish the peer review history of their article (what does this mean?). If published, this will include your full peer review and any attached files.

Reviewer #1: No

Reviewer #2: **Yes: **Alice Fusaro

Reviewer #3: No

Figure Files:

Data Requirements:

Reproducibility:

References:

---

## [Editor Report · Decision Letter 1]

1 Apr 2022

Dear Dr Hill,

We are pleased to inform you that your manuscript 'Ecological divergence of wild birds drives avian influenza spillover and global spread' has been provisionally accepted for publication in PLOS Pathogens.

Best regards,

Pablo Ramiro Murcia

Guest Editor

PLOS Pathogens

Marco Vignuzzi

Section Editor

PLOS Pathogens

Kasturi Haldar

Editor-in-Chief

PLOS Pathogens

orcid.org/0000-0001-5065-158X

Michael Malim

Editor-in-Chief

PLOS Pathogens

orcid.org/0000-0002-7699-2064

I congratulate the authors on the work they performed to address the reviewers' comments. I am satisfied with the way the authors addressed them.
---

## [Editor Report · Acceptance letter]

26 Apr 2022

Dear Dr Hill,

We are delighted to inform you that your manuscript, "Ecological divergence of wild birds drives avian influenza spillover and global spread," has been formally accepted for publication in PLOS Pathogens.

Best regards,

Kasturi Haldar

Editor-in-Chief

PLOS Pathogens

orcid.org/0000-0001-5065-158X

Michael Malim

Editor-in-Chief

PLOS Pathogens

orcid.org/0000-0002-7699-2064